# A Novel Two-Gene Expression-Based Prognostic Score in Malignant Pleural Mesothelioma

**DOI:** 10.3390/diagnostics13091556

**Published:** 2023-04-26

**Authors:** Velizar Shivarov, Georgi Blazhev, Angel Yordanov

**Affiliations:** 1Department of Experimental Research, Medical University Pleven, 5800 Pleven, Bulgaria; 2Department of Genetics, Faculty of Biology, St. Kliment Ohridski Sofia University, 1164 Sofia, Bulgaria; 3Department of Gynaecological Oncology, Medical University Pleven, 5800 Pleven, Bulgaria

**Keywords:** gene expression, prognosis, prediction, score, mesothelioma

## Abstract

Background: Malignant pleural mesothelioma (MPM) is a rare cancer type with an increasing incidence worldwide. We aimed to develop a rational gene expression-based prognostic score in MPM using publicly available datasets. Methods: We developed and validated a two-gene prognostic score (2-PS) using three independent publicly available gene expression datasets. The 2-PS was built using the Robust Likelihood-Based Survival Modeling with Microarray Data method. Results: We narrowed down the model building to the analysis of 179 genes, which have been shown previously to be of importance to MPM development. Our statistical approach showed that a model including two genes (GOLT1B and MAD2L1) was the best predictor for overall survival (OS) (*p* < 0.0001). The binary score based on the median of the continuous score stratified the patients into low and high score groups and also showed statistical significance in uni- and multivariate models. The 2-PS was validated using two independent transcriptomic datasets. Furthermore, gene set enrichment analysis using training and validation datasets showed that high score patients had distinct gene expression profiles. Our 2-PS also showed a correlation with the estimated infiltration by some immune cell fractions such as CD8+ T cells and M1/2 macrophages. Finally, 2-PS correlated with sensitivity or resistance to some commonly used chemotherapeutic drugs. Conclusion: This is the first study to demonstrate good performance of only two-gene expression-based prognostic scores in MPM. Our initial approach for features selection allowed for an increased likelihood for the predictive value of the developed score, which we were also able to demonstrate.

## 1. Introduction

Primary malignant pleural mesothelioma (MPM) is a rare malignancy with an aggressive course and a dismal prognosis for 5-year overall survival of less than 10%. It is currently considered a premier example of a chemically-induced cancer due to occupational or environmental factors because the major risk factor remains a history of asbestos exposure. The elucidation of asbestos exposure as a primary risk factor for MPM led to its restricted use worldwide, which resulted in a slight decline in the incidence of MPM in developed Western countries. However, most of the cases due to occupational exposure develop with a latency of approximately 20 years or even longer, which suggests that we may still expect to witness a peak or a steady state in MPM incidence in various regions worldwide beyond 2020 [1]. Histologically, MPM is classified into three major subtypes, namely epithelioid, sarcomatoid, and biphasic subtypes. Histological subtyping has prognostic value, with epithelioid MPM having a much better prognosis in comparison to sarcomatoid and biphasic cases. Recent molecular profiling studies, however, suggested that the clonal structure of MPM is much more complex and actually consists of subpopulations of the two major histological subtypes (epithelioid and sarcomatoid) in a variable proportion. At a genomic level, MPM is characterized by a relatively low mutational burden [2]. The recurrently mutated genes are mainly tumor suppressor genes such as *BAP1*, *CDKNA2*, *NF2*, *SETD2,* and *TP53* [3]. Unfortunately, most patients with MPM present initially with inoperable disease and are assessed for fitness for systemic therapy and symptoms control approaches. Since the mid-2000s, the mainstay of frontline chemotherapy is cisplatin drug in combination with pemetrexed, with some evidence that anti-angiogenic therapy may add benefit to this doublet. Immune checkpoint inhibitors were shown to provide survival benefits in relapsed settings [4] and gained regulatory approval. Phase 3 results also favor immune checkpoint inhibitors in comparison to chemotherapy in first-line settings [5,6].

In the last two decades, a number of studies addressed the development of gene expression-based prognostic scores. Because of the relatively small number of cases in each study and the diverse patient profiles of each study, the wide applicability of each of those is debatable. Additionally, it will be of greater value if any prognostic score has some predictive power as well. Therefore, the MPM field still requires the identification of reliable prognostic or predictive biomarkers. These become easier to address more than ever before because of the recent advances in genomic technologies and the possibilities to integrate multilayer omics data.

Here we aimed for the development of a novel gene expression-based prognostic score in MPM with potential predictive power. We further aimed to integrate of gene expression and epigenetic data to interrogate the underlying biological features of the newly defined prognostic subgroups of the disease. For the purposes of this study, we used, as a training set, the publicly available dataset of MPM cases that had been profiled as part of The Cancer Genome Atlas (TCGA). As validation datasets, we used two other datasets of gene expression data from MPM reported in the last decade. We were able to develop a 2-gene expression-based score, which showed independent prognostic power in both training and validation datasets. Furthermore, the score has some predictive power as demonstrated by the integration of gene expression and drug sensitivity data from MPM cell lines.

## 2. Materials and Methods

### 2.1. Datasets

We identified 3 datasets of whole transcriptome analysis of at least 50 MPM patients performed in the last decade as outlined in a recent review [7]. As a training dataset, we used RNA-Seq data for 87 MPM cases from the TCGA project [3]. TCGA data were downloaded as RPKM and log10 transformed from the Genomics Data Commons Portal (https://portal.gdc.cancer.gov/ (accessed on 25 March 2023)). As validation datasets, we used data from one study using RNA sequencing (Bueno) [8] of 211 MPM cases and one study with a planar gene expression array (Blum) with 67 MPM cases [9]. Bueno RNA-seq data were downloaded as raw counts from the European Genome-phenome Archive (EGA) from a study, EGAS000010015631, conducted by Bueno et al. (EGAD00001001915, *n* = 211) sequenced via Illumina HiSeq 2000 technology. Paired-end reads were trimmed using Trim Galore v. 0.6.3 to remove Illumina adapters. Paired-end fastq files were subjected to a quality control procedure using FastQC v. 0.72. fastq files were then to reference the human genome hg38 build using HISAT2 v. 2.1 and subsequently annotated to the gene level using featureCounts v 1.6.4. Raw counts data were subsequently voom transformed and quantile normalized using *limma* package for R. Blum dataset data were downloaded as raw .cel files from the Array Express server with the accession number E-MTAB-6877. They were RMA transformed and quantile normalized using *limma* package and analyzed as described below. The overall analytical approach is summarized in Figure 1.

### 2.2. Model Building

In order to reduce the number of features (genes) that we start with in the model build, we decided to select only genes on which MPM cells lines have already been reported to be dependent as part of the DepMap project (https://depmap.org/portal/ (accessed on 25 March 2023)) [10,11,12,13]. We identified a total of 179 genes that were reported to confer significant dependence of MPM in at least one RNA interference screen included in the DepMap project (Appendix A). We subsequently used RNA-Seq data for 87 MPM cases included in the TCGA project as a training dataset to build a model based on the expression of survival-associated selected genes. We filtered the RPKM expression values of the identified 179 genes and applied the Robust Likelihood-Based Survival Modeling with Microarray Data [14], which was implemented through the *rbsurv* package for the R statistical environment. This technique utilizes the partial likelihood of the Cox model and functions through the generation of multiple gene models. It also divides the input dataset into training and validation sets and performs multiple cross-validations of a series of gene models so that it finally provides the optimal model based on the Akaike Information Criterion (AIC). The AIC measure does not have any biological meaning. It is interpreted as a statistical criterion to select the best model. The lower the value of the criterion, the better the statistical model performs for the given dataset. The *rbsurv* package automatically selects and proposes the best multivariate model with the lowest AIC. Cox regression coefficients for the genes included in the model for both datasets were obtained using the *survival* package for R. A total continuous score was calculated for each sample through weighted summation of the gene expression values according to the formula: Scorei=∑wj∗xij where xij is the log-transformed expression value for the gene j in patient i, and wj is the weight assigned to probe set j (here wj was the Cox regression coefficient from the univariate analysis in the training set). The total score was calculated for each patient sample in the training and validation datasets. To build a binary score (high vs. low), we defined a cut-off value specific for each cohort corresponding to the median of the continuous score for the respective cohort. Performance of the median cut-off score was assessed using Receiver Operating Characteristics (ROC) curves implemented through the web-based graphical user interface of the Cutoff Finder package for R (https://molpathoheidelberg.shinyapps.io/CutoffFinder_v1/ (accessed on 25 March 2023)) [15]. Univariate and multivariate analyses for correlation of the continuous and discrete scores with overall survival were performed for each of the datasets using the *survminer* package for R. We have previously used an identical approach for the development of a microRNA-expression based prognostic score in acute myeloid leukemia (AML) [16].

### 2.3. Gene Set Enrichment Analysis

Gene set enrichment analysis (GSEA) was performed using the stand-alone version GSEA 4.0.3 (Boston, MA, USA) developed by Broad Institute (http://www.broadinstitute.org/gsea/index.jsp (accessed on 25 March 2023)) [17]. The cancer hallmarks collection of oncogenic gene ontology signatures from the Molecular Signatures Database (MSigDB) (https://www.gsea-msigdb.org/gsea/msigdb/human/collections.jsp#H (accessed on 25 March 2023)) was used as the reference gene-sets to test for enrichment in the gene expression profiles of high score versus low score patients from each cohort.

### 2.4. Cibersort

We downloaded the CIBERSORT-inferred immune signatures [18] for TCGA mesothelioma cases, which were determined as part of the The Immune Landscape of Cancer study (https://gdc.cancer.gov/about-data/publications/panimmune (accessed on 25 March 2023)) [19]. We used the CIBERSORTx portal (https://cibersortx.stanford.edu/index.php (accessed on 25 March 2023)) [20] to estimate the immune cell fractions of the samples from the Bueno dataset. Pearson correlations between continuous score and each immune cell fraction were calculated using the *cor* function from the stats package for R. Estimated correlation coefficients were presented graphically using the *corplot* package for R.

### 2.5. Drug Sensitivity Analysis

RNA-Seq data for 16 MPM cells lines included in the Sanger Genomics of Drug Sensitivity in Cancer Project (GDSC) [21,22] were downloaded from the ArrayExpress server with the accession number E-MTAB-3983 (https://www.ebi.ac.uk/biostudies/arrayexpress/studies/E-MTAB-3983 (accessed on 25 March 2023)). FRPKM gene expression values were log2 transformed and the 2-PS was calculated for each MPM cell line using the regression coefficient as described above. The GDSC1 analysis dataset included data for 16 MPM cell lines treated with 345 unique compounds. The GDSC2 analysis dataset included data for 15 MPM cell lines treated with 175 drugs. We analyzed the correlation between the 2-PS of the analyzed cell lines and the Area Under the Curve (AUC) of the dose–response analysis of each of the drugs in the two datasets. ANOVA models were used to identify significant correlations, with a two-sided *p*-value of 0.05 being considered significant.

### 2.6. Common Statistical Procedures

All statistical procedures were performed using the R v. 4.2.2 environment for statistical computing. The chi-squared test was used for the assessment of independence in the distribution of categorical variables. A two-sided *t*-test for independent samples was used to compare the means of normally distributed continuous variables. The Wilcoxon–Mann–Whitney test was used to compare the medians of continuous variables without normal distribution. For all statistical tests, an alpha level of 0.05 was considered statistically significant.

## 3. Results

### 3.1. Building and Initial Performance of a Two-Gene Prognostic Score (2-PS)

We applied the Robust Likelihood-Based Survival Modeling with Microarray Data to the training dataset (TCGA dataset) with genes that MPM cell lines were shown to be dependent on (Appendix A) (Figure 1). We chose such an approach for two main reasons. Firstly, we were able to reduce the number of features (genes) to be tested in the prognostic model. Secondly, this approach, in our view, increases the likelihood that the prognostic model built may also have some predictive power. The algorithm selected a best performing prognostic model with AIC of consisting of two genes, *GOLT1B* and *MAD2L1*. The estimated Cox regression coefficients (ln(HR)) for *GOLT1B* and *MAD2L1* were 1.403 and 0.945, respectively. The continuous score for each sample in each dataset was calculated as the sum of expression values for each of the genes in the model multiplied by the regression coefficient. In univariate analysis, the continuous score was prognostic for the OS (Cox regression *p* = 9.69 × 10^−10^). We further defined a binary score using as a cut-off the median of the continuous score for all samples. In univariate analysis, the binary score also showed significant prognostic value (Cox regression *p* = 2.85 × 10^−6^ and Figure 2A) with an Area Under the Curve value of the Receiver Operator Characteristics (ROC) analysis of 0.67 (Figure 2B). This AUC value is distinct from 0.5 suggesting there exists a true difference in survival between the two groups of patients [23]. As it is below 0.7, it is expected that the binary score may not be the only contributing prognostic factor in this cohort. Therefore, we further evaluated the performance of the binary score in a multivariate model with age, sex, stage, histology, and mutational status as covariates. Notably, the binary score retained independent prognostic power (Cox regression *p* = 7.34 × 10^−6^, Figure 2C).

### 3.2. Validation of the 2-PS

In order to validate our 2-gene prognostic score, we used two recent publicly available datasets with either RNA-Seq data (*n* = 211) (Bueno) or planar expression arrays (*n* = 67) (Blum). The estimated continuous score in the Bueno dataset showed a clear prognostic value (Cox regression *p* = 1.22 × 10^−7^). The score was further converted to a binary one using the median of the continuous score as a cut-off. Analogous to the data training dataset, the binary score in this validation dataset also had prognostic power (Cox regression *p* = 1.4 × 10^−5^, Figure 3A) with an AUC of the ROC analysis of 0.75 (Figure 3B), which is considered acceptable for a diagnostic test [23]. Furthermore, in the extensive multivariate model for the Bueno dataset, the binary score was still of independent prognostic value (Cox regression *p* = 0.03, Figure 3C).

Following the same procedure, we analyzed the performance of the estimated continuous score in the Blum dataset. In multivariate analysis, the continuous model was a significant prognostic factor (*p* = 1.13 × 10^−5^). The same also held true for the binary model defined by the cut-off of the median for the continuous score (Cox regression *p* = 0.0014, Figure 4A). The AUC in the ROC analysis, in that case, was 0.85 (Figure 4B), which is considered excellent performance for a diagnostic test [23]. Finally, a multivariate model for the Blum dataset was built using binary score, sex, age, stage, and histology, albeit without mutational data, as those were not publicly available. The model demonstrated the independent prognostic value of the binary score (Cox regression *p* = 0.00671, Figure 4C).

### 3.3. Gene Set Enrichment Analysis

Based on the observation that our novel 2-PS performed similarly well in both training and the two validation datasets, we hypothesized that the score may correlate with specific gene expression signatures. Therefore, we performed GSEA using predefined cancer hallmark signatures from the MSig database. For each of the datasets, we obtained a number of signatures enriched in the high score patients’ subgroups as follows: TCGA (*n* = 37), Bueno (*n* = 34), Blum (*n* = 34). There was a more significant overlap between the overexpressed signatures in the TCGA and Bueno datasets and slightly less so between any of those two and the Blum dataset. However, there were a total of 25 signatures that were commonly overexpressed in high score patients from the three cohorts (Figure 5A). Most of them were related to DNA repair and DNA damage response (Figure 5B).

### 3.4. Correlation with Immune Signatures

Infiltrating immune cells are a major player in the immune response against cancer and may be used as prognostic and predictive markers. We therefore questioned whether 2-PS-defined groups of MPM patients would also have distinct underlying profiles in terms of the microenvironment. To address this question, we analyzed whether the 2-PS correlated with specific immune cell subtype infiltration in MPM. We used the inferred infiltrating immune cell fractions using the CIBERSORT algorithm using TCGA (Figure 6A) and Bueno (Figure 6B) datasets. Notably for both datasets the continuous prognostic score showed a positive correlation with CD8+ T cell fraction as well as with M1 and M2 macrophage fractions.

### 3.5. Potential Predictive Power of the 2-PS

We specifically developed our score focusing on genes for which there had been a demonstration of dependency using some knock-out screen experiments. It was therefore reasonable to accept that the score may have some potential predictive power. The most straightforward approach to provide some preliminary evidence in that regard was to use drug sensitivity screens data from MPM cell lines. We calculated the 2-gene prognostic score for each of the cell MPM cell lines included in the Genomics of Drug Sensitivity in Cancer project GDSC1 part (*n* = 16) and GDSC2 part (*n* = 15) [21,22]. We subsequently tested the correlation between the 2-PS and the sensitivity to each of the drugs tested in both projects using AUC values. The AUC-defined response to 11 drugs from the GDSC1 set showed a significant correlation with the 2-PS of the tested mesothelioma cell lines (Figure 7A); whereas for the GDSC2 set, the number of such significant correlations was 18 (Figure 7B). Interestingly, this analysis revealed a correlation of the 2-PS with response to commonly used drugs in mesothelioma management such as cisplatin (R = −0.51, *p* = 0.046), gemcitabine (R = 0.69, *p* = 0.019), and vinblastine (R = 0.63, *p* = 0.037).

## 4. Discussion

A number of studies proposed gene expression-based prognostic models in MPM [3,8,9,24,25,26,27,28,29,30,31]. They differ significantly in their approaches for feature selection, training and validation datasets, the number of genes included in the final model, as well as in the performance in different MPM cohorts. Additionally, the predictive value of each proposed score remains largely unexplored. Here we implemented a novel approach in features selection by limiting the number of genes to be tested for inclusion in an MPM prognostic model only to genes for which it has been that MPM cell lines have been sensitive to their knock-down. We then applied the RBSURV approach to the TCGA dataset and built a two-gene prognostic model, which showed moderate prognostic power as a continuous or binary score in both univariate and multivariate models in three different MPM cohorts. This moderate prognostic power is in trade-off with the minimal number of genes included in the prognostic score, avoiding over-fitting of the model by the inclusion of a higher number of features. The limited number of genes in our 2-PS may further allow simple validation using low-throughput techniques such as quantitative RT-PCR or immunohistochemistry. We tested the performance on continuous and binary scores using univariate and multivariate Cox regression analyses in three independent cohorts (two from the USA and one from Europe). The definition of cut-offs for binary scores cannot be directly applied to prospective studies as we used a dataset-specific median to define binary scores provided that the datasets we used were from different gene expression profiling platforms and had different pre-processing steps. Ideally, the model must be validated in a prospective fashion using a simple, readily reproducible technique providing a uniform standardized read-out of gene expression. The two genes included in our model are not widely studied in MPM.

The *MAD2L1* gene encodes the mitotic arrest deficient 2 like 1, coding for the respective protein, which is an integral part of the mitotic spindle assembly checkpoint and ensures that all chromosomes are properly aligned at the metaphase plate before the cell can proceed to anaphase [32]. MAD2L1 was recently found overexpressed in several MPM cell lines at the mRNA and protein level [33]. This study conforms with a previous one which demonstrated higher MAD2L1 protein expression (both nuclear and cytoplasmic) in MPM cell lines as compared to normal mesothelium [34]. Interestingly, according to the latter study, the total *MAD2L1* mRNA expression level did not correlate with the overall survival of 80 MPM patients [34]. However, the same study showed that higher nuclear MAD2L1 expression determined using immunohistochemistry correlated with a shorter overall survival [34]. Notably, a recent study showed that BRCA1 in the mesothelioma leads to the co-depletion of MAD2L1 mRNA and protein [35]. Additionally, loss of BRCA1/MAD2L1 was associated with resistance to vinorelbine ex vivo, and the survival was shorter for patients lacking BRCA1/MAD2L1 expression in comparison to those with double-positive tumors [35]. This observation can explain the fact that our 2-PS correlated with resistance to vinblastine (mitotic spindle assembly inhibitor) and olaparib (PARP inhibitor) in MPM cell lines (Figure 7). In addition, among the top enriched pathways in the GSEA analysis of all three cohorts were pathways directly involving mechanisms of DNA replication such as the following pathways: “Mitotic spindle”, “G2M checkpoint”, and “DNA repair” (Figure 5).

The *GOLT1B* gene encodes for the human vesicle transport protein (Golgi Transport 1B) GOT1B protein [36]. GOLT1B might be overexpressed in various tumors because of the amplification of the chromosome 12p region [37]. Recent studies show that overexpression of GOLT1B in breast and colorectal cancer might be associated with poorer outcomes due to the promotion of immune evasion [38,39]. Consistent with that, we found that in high 2-PS MPM, patients from all three cohorts in our analysis, “Epithelial mesenchymal transition”, “Apical junction”, and “Protein secretion pathways”, were significantly enriched in the gene expression profiles (Figure 5).

Recent reports regarding the role of GOLT1B in immune evasion let us investigate whether our 2-PS correlated with the estimated fractions of immune cells within the tumor tissue. We used the now standard deconvolution algorithm to obtain those fractions and were able to demonstrate that 2-PS correlated with the CD8+ T cells and M1/2 macrophage content (Figure 6). Using a similar approach, Blum et al. demonstrated that epithelioid-like morphology and transcriptomic profile correlated with an estimated fraction of CD8+ T cells [9]. Nguen et al. also demonstrated that inferred infiltrating immune cell fractions can be combined with genomic parameters to develop prognostic models in MPM [40]. Finally, another recent study showed that markers for higher levels of systemic inflammation correlated with shorter overall survival in MPM patients [41]. Our observations in the context of those studies obviously suggest that immune-based markers are to be included in prognostic schemes for MPM patients. In addition, it is rational to expect that they may have predictive power for the success of immune-checkpoint inhibitors (ICIs)-based therapy in MPMs [4,6].

However, even in the era of ICIs, combinations with conventional chemotherapy or targeted therapy may yield additional clinical benefits in MPM. Therefore, we further evaluated our 2-PS as a possible marker to predict the sensitivity of MPM cell lines to small molecule drugs. Interestingly, 2-PS inversely correlated with AUC values for cisplatin, suggesting that it may predict higher sensitivity to cisplatin. The opposite observation was made for two other common chemotherapeutics, gemcitabine and vinblastine, suggesting that our 2-PS can predict resistance to those two. These findings also suggest that the performance of the 2-PS in various datasets might be highly dependent on the therapeutic approach used in any cohort and its further evaluation need to focus on uniformly treated MPM patients.

## 5. Conclusions

In sum, here we demonstrated the development of a 2-gene expression-based prognostic score in MPM with initial filtration of features based on predicted gene dependency of MPM. Our score was further validated in two independent cohorts. Furthermore, it obviously defines patient subgroups with specific gene profile expressions, underlying immune surveillance mechanisms, and drug sensitivity. Our 2-PS can be tested in a prospective fashion using readily available pathological techniques such as RT-PCR and immunohistochemistry. Finally, our approach to the development of the 2-PS can be applied to other cancer types.

## Figures and Tables

**Figure 1 diagnostics-13-01556-f001:**
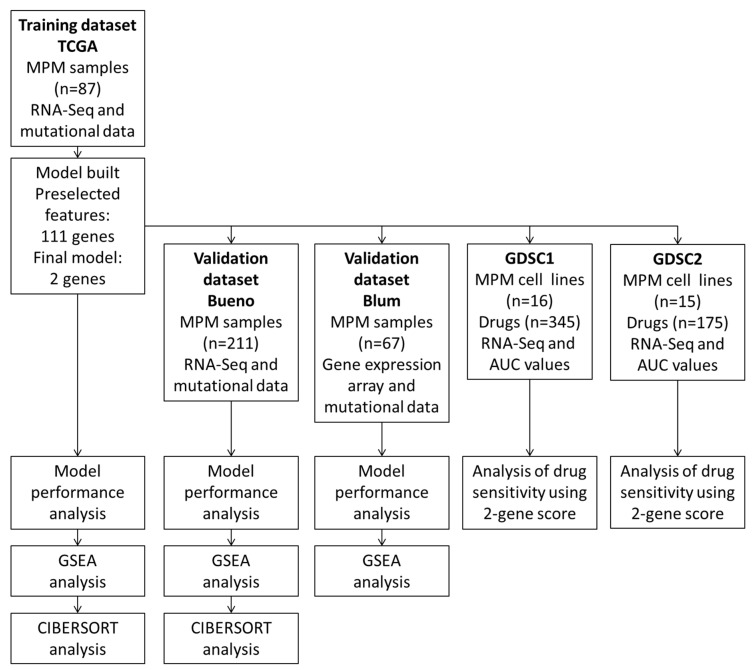
Summary of the analytical approach in the study using different datasets.

**Figure 2 diagnostics-13-01556-f002:**
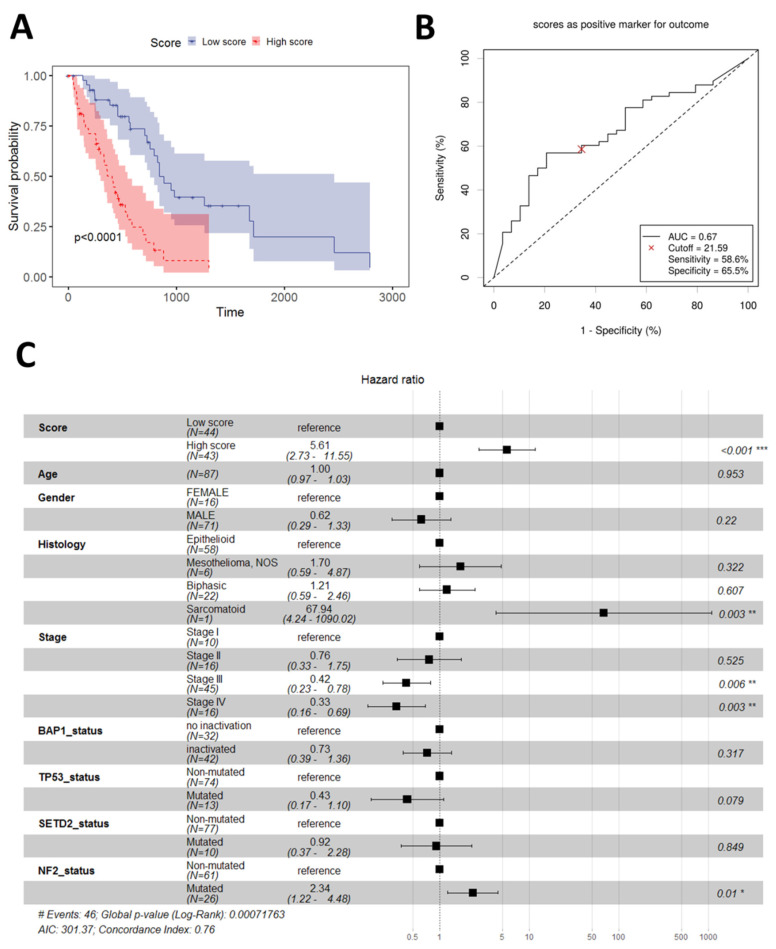
Performance of the 2-gene PS in the TCGA cohort. (**A**) Overall survival in low vs. high score patients; (**B**) ROC analysis using the median value of the continuous score to define low and high score patients; (**C**) Multivariate prognostic model including 2-PS and other demographic and clinical parameters. Significance notations: “*”—<0.05, “**”—<0.01, ”***”—<0.001.

**Figure 3 diagnostics-13-01556-f003:**
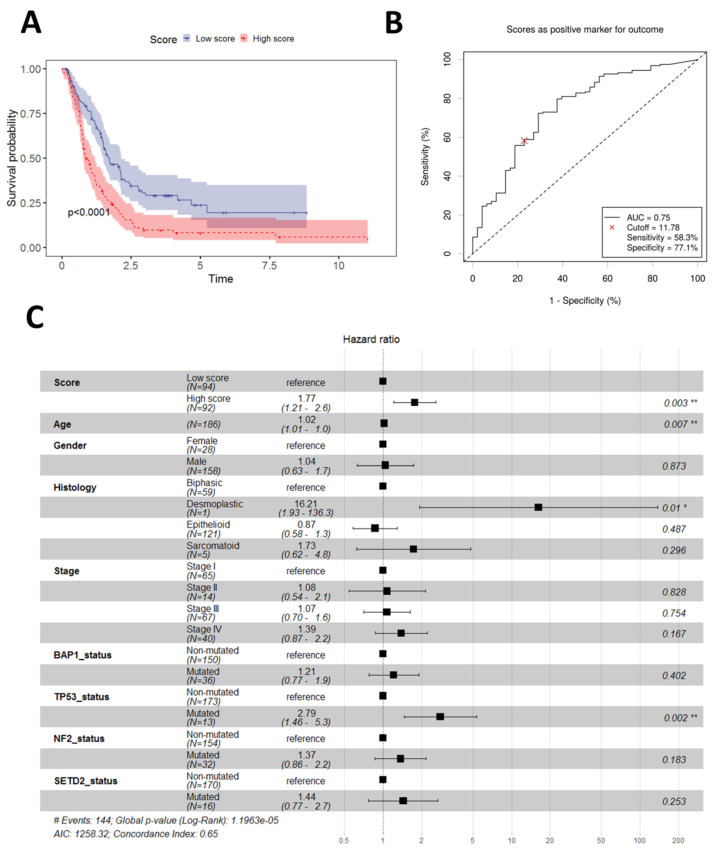
Performance of the 2-gene PS in the Bueno cohort. (**A**) Overall survival in low vs. high score patients; (**B**) ROC analysis using the median value of the continuous score to define low and high score patients; (**C**) Multivariate prognostic model including 2-PS and other demographic and clinical parameters. Significance notations: “*”—<0.05, “**”—<0.01.

**Figure 4 diagnostics-13-01556-f004:**
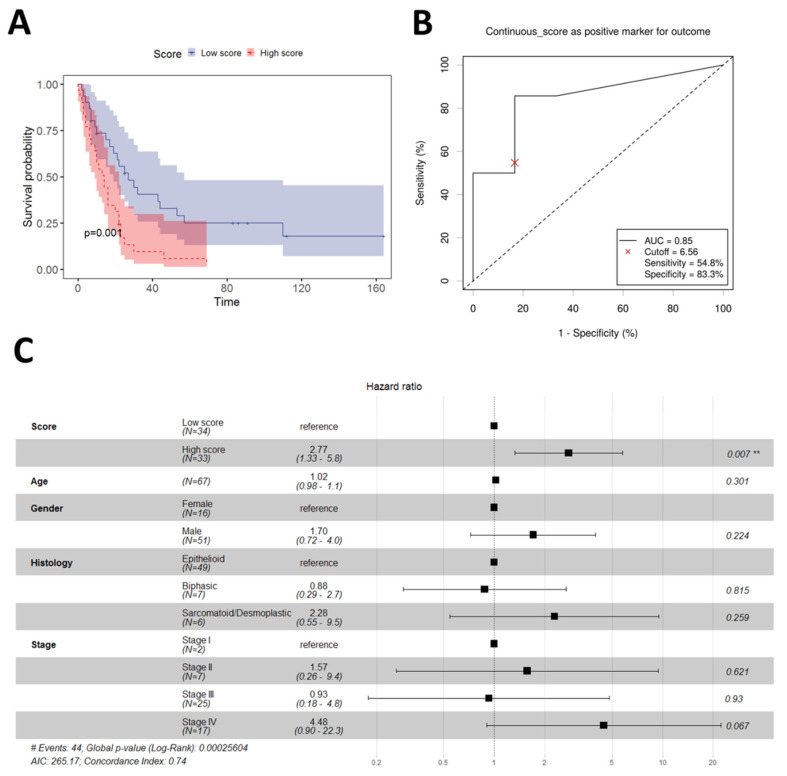
Performance of the 2-gene PS in the Blum cohort. (**A**) Overall survival in low vs. high score patients; (**B**) ROC analysis using the median value of the continuous score to define low and high score patients; (**C**) Multivariate prognostic model including 2-PS and other demographic and clinical parameters. Significance notations: “**”—<0.01.

**Figure 5 diagnostics-13-01556-f005:**
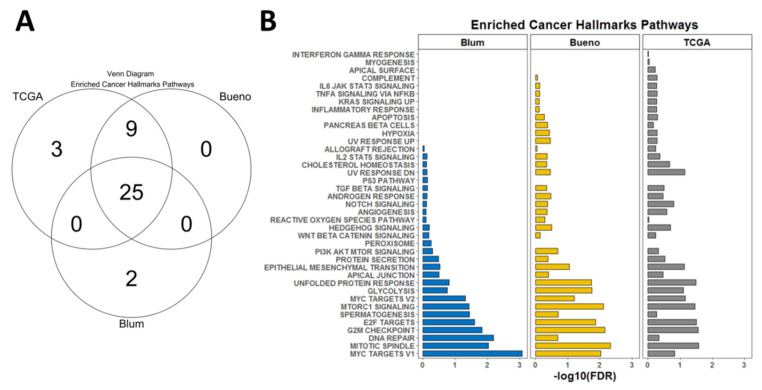
Summary of the GSEA analysis in all three cohorts. (**A**) Venn diagram showing the overlap of enriched cancer hallmark pathways in high score patients from the three cohorts. (**B**) List of all enriched cancer hallmark pathways in high score patients from the three cohorts at a false discovery rate (FDR) of below 0.05.

**Figure 6 diagnostics-13-01556-f006:**
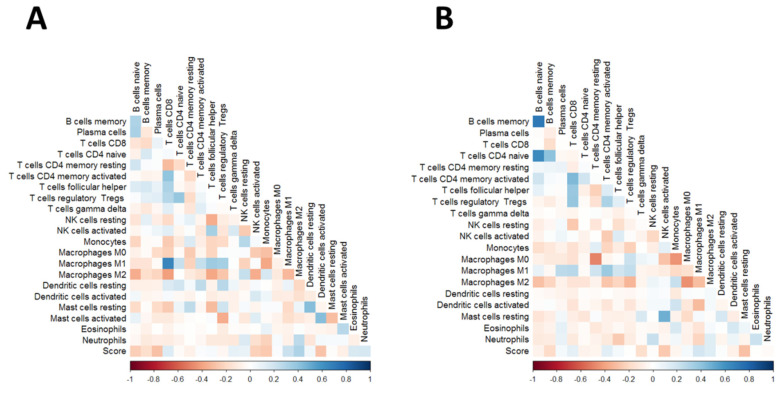
Plots of correlation matrices between 2-gene prognostic score and the estimated immune cell population contents using CIBERSORT. (**A**) TCGA dataset; (**B**) Bueno dataset. Color codes represent Pearson correlation coefficients.

**Figure 7 diagnostics-13-01556-f007:**
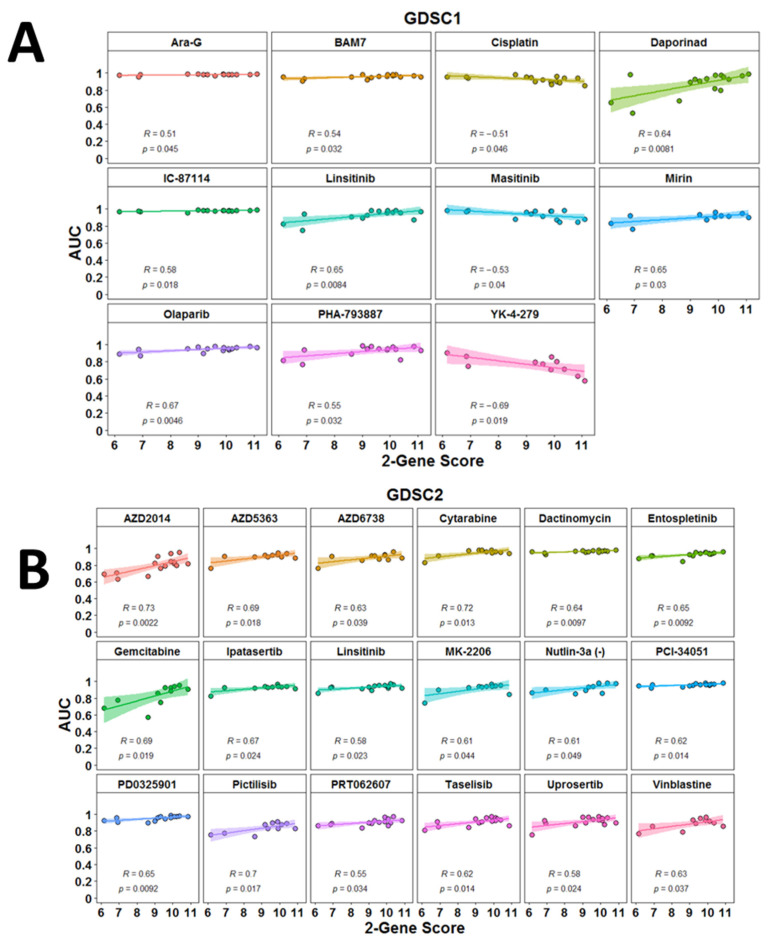
Linear correlations between 2-gene prognostic score and sensitivity of MPM cell lines (as measured by Area Under the Curve (AUC)) to different compounds. (**A**) Data from the analysis using the GDSC1 dataset. (**B**) Data from the analysis using the GDSC2 dataset. Correlation coefficients are from Pearson correlation.

## Data Availability

All data used in this study are publicly available and the sources are referenced in the main text accordingly.

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
