# Peer review of "A Novel Two-Gene Expression-Based Prognostic Score in Malignant Pleural Mesothelioma"

_diagnostics, 2023, doi:10.3390/diagnostics13091556_

Round 1

Reviewer 1 Report

In this study, Velizar et al., developed a 2-gene expression prognostic signature for maligant pleural mesothelioma. Moreover, authors also analyzed the GESA, immune cell infiltration, and drug sensitivity between high and low risk group. The paper was written in a clear and logical manner, and the analysis and discussion are thorough. However, from the personal viewpoint of the reviewers, there are 2 major concerns need to be addressed before this manuscript can be considered for acceptance.

1, It seems that the 2-gene expression prognostic signature genes were selected from 111 genes which were obtained from knockout experiments of the MPM cell lines. The marker gene does not necessarily need to be an essential gene for the MPM cell line. What is the rationale behind this? Please explain.

2, When using the 2-gene model for prediction, the authors used the median of the predicted scores for each test data set as the threshold.  Is this thresholding method reasonable as a predictive model for disease? In actual clinical practice, in most cases, the test data may have only one sample.

Author Response

1, It seems that the 2-gene expression prognostic signature genes were selected from 111 genes which were obtained from knockout experiments of the MPM cell lines. The marker gene does not necessarily need to be an essential gene for the MPM cell line. What is the rationale behind this? Please explain.

Response:

This approach eases the model build by limiting the number of features to be tested. On the other hand it increase the likelihood that the prognostic model would have some reasonable performance as a predictive marker for response to some therapies. That is what we actually demonstrated with this study.

We included the following text:

We chose such an approach for two main reasons. Firstly, we were able to reduce the number of features (genes) to be tested in the prognostic model. Secondly, this approach in our view increases the likelihood that the prognostic model built may also have some predictive power.

2, When using the 2-gene model for prediction, the authors used the median of the predicted scores for each test data set as the threshold.  Is this thresholding method reasonable as a predictive model for disease? In actual clinical practice, in most cases, the test data may have only one sample.

Response:

That is true. Ideally the model must be validated in prospective fashion using technique such as RT-PCR.

We included the following text:

The definition of cut-offs for binary scores cannot be directly applied to prospective studies as we used data-set specific median to define binary scores provided that the data-sets we used were from different gene expression profiling platforms and had different pre-processing steps. Ideally, the model must be validated in a prospective fashion using a simple readily reproducible technique providing uniform read out of gene-expression.

Reviewer 2 Report

Shivnov et.al., propose two interesting genes as progonstic markers in rare cancer malignant pleural mesothelioma. It is as interesting study I have following comments

1)    In figure 2B AUC is only 0.6 it is very low? What is the statistical significance? Authors should explain what is the biological importance of AIC score.

2)    In figure 5 authors present the GSEA analysis is it done based two genes? Or all genes from the gene expression dataasets?

3)    In figure6 why the correlation analysis between immune signature is done? Is there any previous evidence that these two genes are some how related to immune system?

4)    In Figure7 authors performed drug sensitivity analysis they should explain motivation of this analysis does it some how suggests therapeutic approach against genes ? 

Author Response

  • In figure 2B AUC is only 0.6 it is very low? What is the statistical significance? Authors should explain what is the biological importance of AIC score.

Response:

The AUC value is 0.67, which is acceptable performance for a time-dependent ROC analysis. We included the following text:

This AUC value is distinct from 0.5 suggesting there exists a true difference in survival between the two groups of patients (22) as it is below 0.7 it is expected that the binary score may not be the only contributing prognostic factor in this cohort. Therefore, we further…

Regarding AIC we included the following text:

We included the following clarification text in the Methods section: AIC measure does not have any biological meaning. It is interpreted as a statistical criterion to select the best model. The lower the value of the criterion is the better the statistical model performs for the given data set. The rbsurv package automatically selects and proposes the best multivariate model with the lowest AIC.

  • In figure 5 authors present the GSEA analysis is it done based two genes? Or all genes from the gene expression dataasets?

Response:

All genes in a given dataset were used for GSEA analysis.

  • In figure6 why the correlation analysis between immune signature is done? Is there any previous evidence that these two genes are some how related to immune system?

Response:

Biology of the disease is not only heralded by the intrinsic molecular features such as gene expression profile but also by the microenvironment. The purpose of this analysis was to demonstrate that our score defines distinct subtypes of disease also in terms of tumor microenvironment. There is some data that COLT1B gene can be associated with the evasion of immune surveillance in CRC. This is already mentioned in the discussion section.

We added the following text:

We therefore questioned whether 2-PS defined groups of MPM patients would also have distinct underlying profile in terms of the microenvironment. To address this question we…

  • In Figure7 authors performed drug sensitivity analysis they should explain motivation of this analysis does it some how suggests therapeutic approach against genes ? 

Response:

We specifically developed our score focusing on genes which are potentially targets for therapy in MPM as there has been demonstrated that MPM samples were sensitive to their knock-out. It was therefore reasonable to accept that the score may have some potential predictive power. The easiest way to provide some preliminary evidence in that regards is to use drug sensitivity screens data. The identified positive and negative associations does not suggest direct targeting of the two genes in the score but rather demonstrate it potential predictive power for response to some conventional therapies such as platin-based regimens. It also suggests that the performance of the score in various data-sets might be highly dependent on the therapeutic approach used in any cohort.

We included the following text:

We specifically developed our score focusing on genes for which there had been demonstration of dependency using from some knock-out screen experiments. It was therefore reasonable to accept that the score may have some potential predictive power. The most straightforward approach to provide some preliminary evidence in that regards was to use drug sensitivity screens data from MPM cell lines.

These findings also suggests that the performance of the 2-PS in various data-sets might be highly dependent on the therapeutic approach used in any cohort and its further evaluation need to focus on uniformly treated MPM patients.

Round 2

Reviewer 1 Report

thanks for the revisions, the scientific content requires no further revision.